# Microwave quantum diode

**Rishabh Upadhyay** [1] ✉, **Dmitry S. Golubev** [1], **Yu-Cheng Chang** [1], **George Thomas** [1,2], **Andrew Guthrie**[1], **Joonas T. Peltonen** [1] **& Jukka P. Pekola**[1]

The fragile nature of quantum circuits is a major bottleneck to scalable quantum applications. Operating at cryogenic temperatures, quantum circuits are highly vulnerable to amplifier backaction and external noise. Non-reciprocal microwave devices such as circulators and isolators are used for this purpose. These devices have a considerable footprint in cryostats, limiting the scalability of quantum circuits. As a proof-of-concept, here we report a compact microwave diode architecture, which exploits the non-linearity of a superconducting flux qubit. At the qubit degeneracy point we experimentally demonstrate a significant difference between the power levels transmitted in opposite directions. The observations align with the proposed theoretical model. At − 99 dBm input power, and near the qubit-resonator avoided crossing region, we report the transmission rectification ratio exceeding 90% for a 50 MHz wide frequency range from 6.81 GHz to 6.86 GHz, and over 60% for the 250 MHz range from 6.67 GHz to 6.91 GHz. The presented architecture is compact, and easily scalable towards multiple readout channels, potentially opening up diverse opportunities in quantum information, microwave read-out and optomechanics.

Quantum engineering, a dynamic discipline bridging the fundamentals of quantum mechanics and established engineering fields has developed significantly in the past few decades. Two-level systems such as superconducting quantum bits are the building blocks of quantum circuits. Qubits of this type are currently the most researched and used in quantum computing applications[1–5]. The characteristics of the superconducting qubits such as eigen energies, non-linearity, coupling strengths etc. can be tailored easily by adjusting the design parameters[6,7]. Qubits have large non-linearity, which makes it possible to selectively address and control them[1,3,7,8]. This dynamic property makes superconducting qubits a strong candidate for plethora of applications. Other two-level microscopic quantum systems[9–14] also have certain advantages and may be used in the future.

Quantum devices operate at low temperatures and require good isolation from external noises. Microwave devices, such as circulators and isolators, protect quantum circuits by unidirectionally routing the output signal, whilst simultaneously isolating noise from the output channel back to the quantum circuit. Their non-reciprocal character relies on the properties of ferrites[15–17]. Ferrite-based non-reciprocal devices are bulky[15–17], and they cannot be positioned near the quantum

circuit because they require strong magnetic fields. Although commercial ferrite based non-reciprocal devices harness high isolation and low insertion loss, their dependency on magnetic components limits the scalability of cryogenic quantum circuits[15,16,18,19]. Various ferrite-free approaches based on non-linear behavior of artificial atoms[16], dc superconducting quantum interference devices (dc-SQUID)[20,21], and arrays of Josephson junctions (JJ's)[19,22–24], have been experimentally demonstrated and implemented. Recently, a circuit based on semiconductor mixers has been used to realize a compact microwave isolator, which the authors claim could be extended to an on-chip device using Josephson mixers, although the "on-chip" demonstration is not yet reported[25]. Additionally, mesoscopic circulators exploiting the quantum Hall effect to break time-reversal symmetry of electrical transport in 2D systems are explored at a cost of larger magnetic fields deleterious to superconducting circuits[18,26–29]. More recently, a passive on-chip circulator based on three Josephson elements operating in charge-sensitive regime was demonstrated[30]. Such devices are frequently limited by their parameter regime, leaving them charge sensitive and therefore difficult to implement in a practical scenario. However, it is possible to mitigate the charge-sensitivity by carefully

[1]Pico group, QTF Centre of Excellence, Department of Applied Physics, Aalto University School of Science, P.O. Box 13500, 00076 Aalto, Finland. [2]VTT Technical Research Centre of Finland Ltd, Tietotie 3, 02150 Espoo, Finland. ✉e-mail: rishabh.upadhyay@aalto.fi

tuning the device parameters. Our device operates in a parameter regime that is not sensitive to charge fluctuations or charge parity switching, a fundamental requirement for any practical implementation, and requires small magnetic field. The reported device is a proof of concept (PoC), potentially useful in the applications relevant to microwave read-out components in the field of superconducting quantum circuits.

In this work, we present a robust and simple on-chip microwave diode demonstrating transmission rectification based on a superconducting flux qubit[8]. The concept of the device is shown in Fig. 1a. The flux qubit is inductively coupled to two superconducting resonators of different lengths with different coupling strengths. The design details are reported later in this section. Probing the qubit at the half-flux (degeneracy point) with one tone-spectroscopy, we observe identical patterns of transmission coefficient for signals propagating in

the opposite directions, which are shifted by 5 dB in power. This shift indicates the non-reciprocal behaviour in our device, expressed in terms of transmission rectification ratio (R) in this article. The origin of this effect is the non-linearity of the flux qubit, which controls the transmission coefficient of the whole structure. Since the qubit is asymmetrically coupled to the input ports 1 and 2, via length dependent inductive couplers and asymmetric resonators, different powers should be applied through these ports to excite the qubit to the same level. We study the transmission rectification ratio, $R$, under different injected microwave powers and in a wide range of frequencies and magnetic fluxes applied to the qubit. Due to its strong non-reciprocity, our device could potentially be utilized as a ferrite free on-chip isolator in a microwave readout scheme[18–20,24]. The strong non-reciprocal behaviour observed in the reported device is relevant in the field of circuit quantum thermodynamics (c-QTD)[31,32] to facilitate and manage

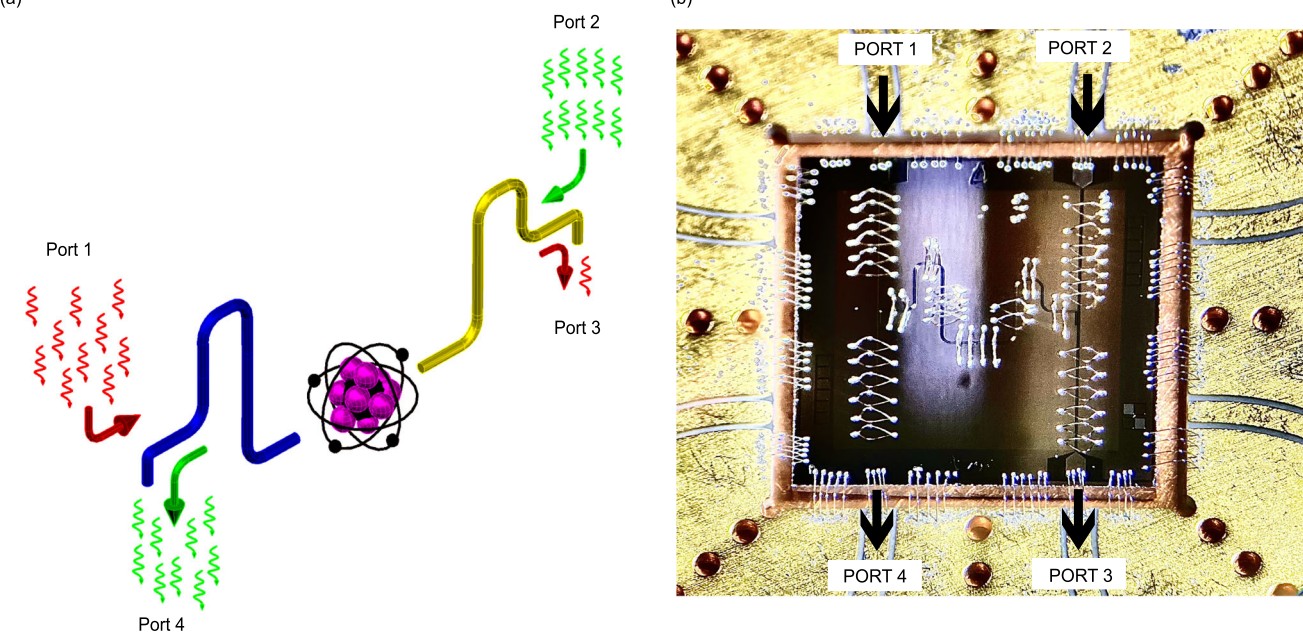

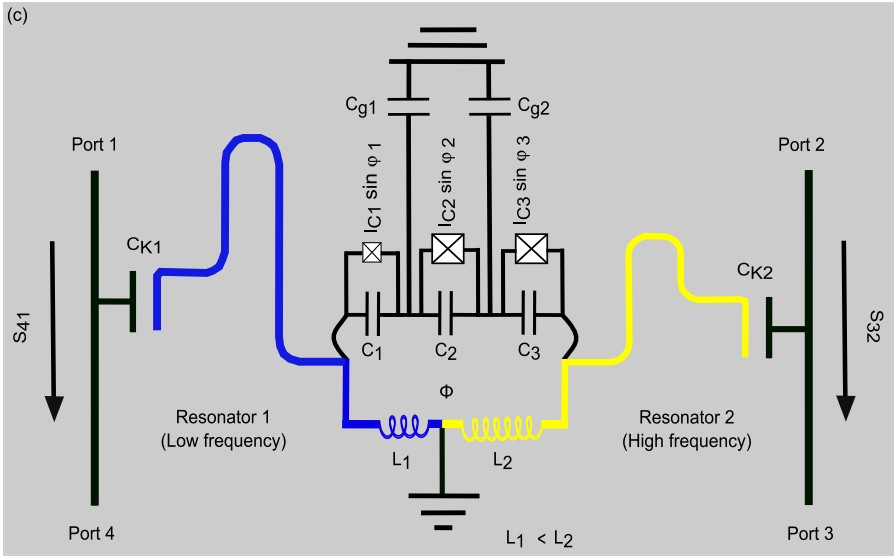

**Fig. 1 | The studied device. a** Conceptual representation of an artificial atom coupled to two resonators. The photons pass easily from the right side (green arrows, from port 2) to the left side (port 4), whereas those coming from left side (red arrow, from port 1) are mainly reflected back. **b** Optical microscope image of the device bonded on gold plated copper sample stage. The RF signal enters either via port 1 or port 2. Port 3 and Port 4 are the output ports. The black arrows indicate the direction of signal propagation. **c** Circuit model of the device. Here, Φ is the external magnetic flux threading through the qubit loop.

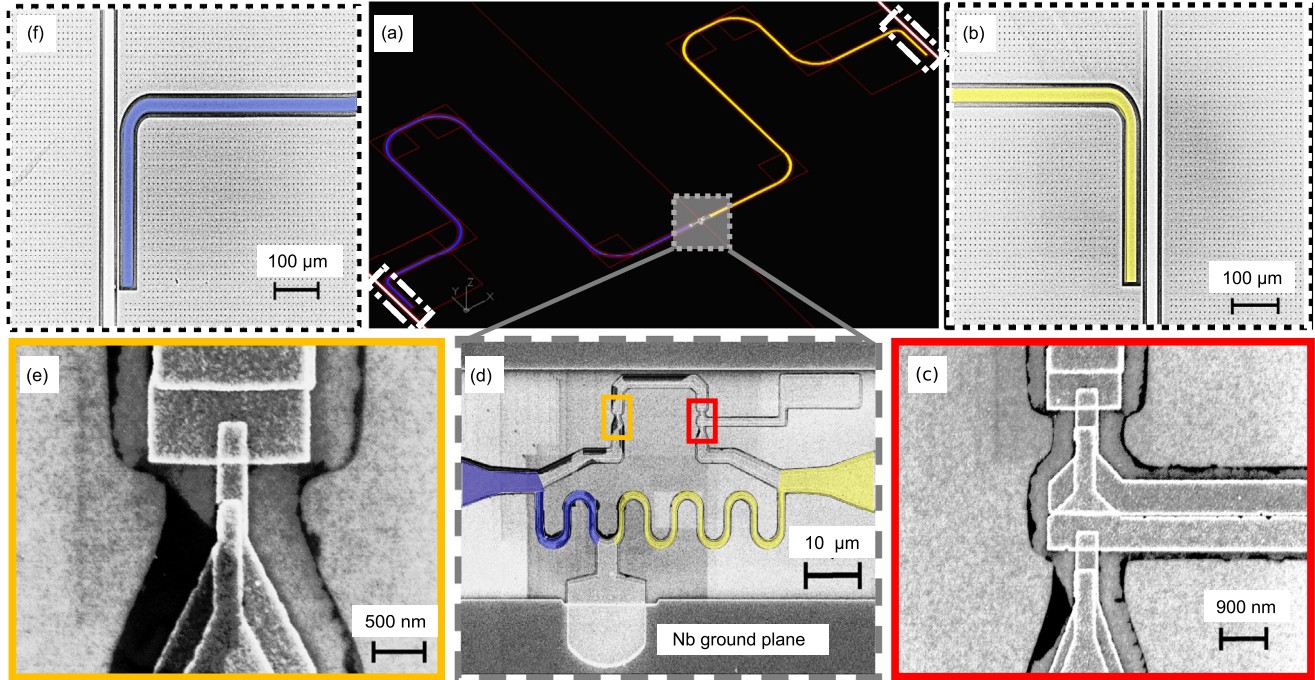

**Fig. 2 | Electron micrographs of the reported device. a** The CAD layout of the device exhibiting the two resonators and the three-junction superconducting flux qubit at the center. The right and the left resonators are shown by yellow and by dark-blue colors respectively. The areas enclosed with white dotted lines at the top-right and bottom-left corner show the locations of the capacitors coupling the right and the left resonators with the feedlines. In (**b, f**) we show the magnified images of these capacitors. The gray shaded area close to the center in (**a**) shows a three-terminal flux qubit. Its zoomed image is shown in (**d**). The qubit is coupled to both resonators via the local inductances to the left and right, highlighted with different colors. **c** An enlarged electron micrograph of the two big junctions of the flux qubit. **e** Electron microscope image of the smaller qubit junction.

the heat flow in superconducting quantum circuits[33–40]. Moreover, its compact size makes it suitable for multiple read-out channels. The possibility to control the transmission rectification ratio with tiny magnetic fields provides an additional advantage.

## Results and discussion

### Design

An optical microscope image of the device is shown in Fig. 1b, and the circuit diagram in Fig. 1c. The core element of the device is a three-junction superconducting flux qubit[8,41]. Its magnified image is presented in Fig. 2d. The superconducting loop of the flux qubit contains Josephson junctions. The areas of junctions numbered as 2 and 3 are nominally the same, whereas the junction 1 is smaller by $\alpha$. The designed junction asymmetry parameter ($\alpha$) is $\approx 0.6$. Hence, the critical currents of junctions 2 and 3 are nominally equal, $I_{C2} = I_{C3} = I_C$, and that of the first junction is given by $I_{C1} = \alpha I_C$ The flux qubit has two superconducting islands, which we number as the island 1 and the island 2. The total capacitance of island 1 is given by $C_{G1} = C_1 + C_2 + C_{g1}$, where $C_1, C_2$ are the capacitances of the junctions 1, 2 and $C_{g1}$ the capacitance to the ground plane. Analogously, the total capacitance of the second island is $C_{G2} = C_2 + C_3 + C_{g2}$. The flux qubit is inductively coupled to the resonators 1 and 2 via the inductances $L_1$ and $L_2$, realized as a superconducting aluminum wire divided in two parts by the grounding electrode, see Fig. 2d. The corresponding coupling constants are proportional to the inductances, $g_j \propto L_j$ ($j = 1, 2$)[42–45]. The left resonator (resonator 1) having the designed frequency $f_1 = 6.5$ GHz, is more weakly coupled to the qubit, while the right resonator (resonator 2), with a designed frequency of $f_2 = 7.5$ GHz, has stronger coupling, i.e. $g_1 < g_2$. These couplings are not experimentally measurable. From our theory model we estimate them as $g_1/(2\pi) = -89$ MHz and $g_2/(2\pi) = 155$ MHz. The frequencies $f_1$ and $f_2$ cited above are the nominal values expected in the limit of $\lambda/4$-resonators, which is achieved at

$L_1, L_2 \to 0$. At finite qubit resonator coupling they may shift downwards by an unknown value. Both resonators are coupled to the transmission lines via nominally equal coupling capacitances as shown in Fig. 2b, f. To probe the system we use two feedlines that are coupled to the right and left resonators with a nominally equal capacitance. One can treat these capacitors as lumped elements since the length of these couplers are much shorter than the wavelength in our interested bandwidth. Therefore the incoming photons from the resonators to port 2 and port 4 would be the same. This can be simply simulated by electromagnetic analysis (SONNET) simulations. Further details of device and its fabrication are reported in Section II of this article.

### Measurement scheme

The device is wire-bonded to a gold-coated printed circuit board as shown in Fig. 1b. The RF spectroscopy measurements have been performed in a cryo-free dilution refrigerator at the base temperature of 15 mK. The measurement setup is schematically shown in Fig. 3a. We have used two input lines with identical attenuation at different temperature stages of the fridge. At the output end, we have used a commercial coaxial microwave switch, which allowed us to connect either port 3 or port 4 of the device to a single output channel. Under applied DC bias this coaxial microwave switch connects the output channel to the desired port and terminates the other port to 50 $\Omega$ to avoid reflections, as shown in Fig. 3b. This setup eliminates unwanted differences in microwave transmission through output ports 3 and 4, which might otherwise affect the measurements. The signal then follows an output line that includes two isolators embedded at the mixing chamber. At the 4 K stage the signal is amplified by a low noise HEMT amplifier by 42 dB. Furthermore, the signal is amplified at room temperature by 52 dB. For the flux bias of the qubit we have used a DC-driven global magnet coil at the mixing chamber, as shown in Fig. 3a. One could also integrate an on-chip proximal current bias lines capable

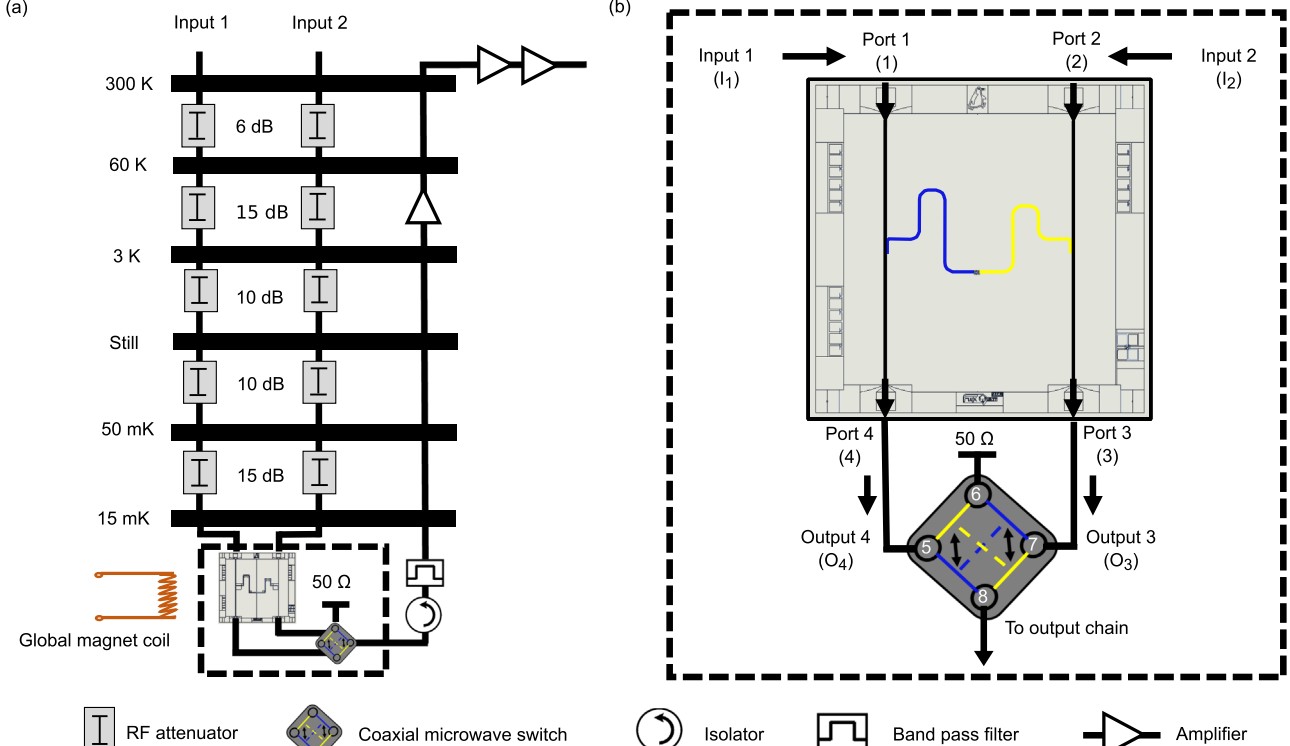

**Fig. 3 | Measurement setup. a** Different temperature stages of the fridge with respective attenuation at each stage. The attenuation in the input lines 1 and 2 are nominally identical. **b** An enlarged image of the sample setup at the mixing chamber. Input 1 connects to the port 1 and input 2 --- to the port 2. The output ports 3 and 4 are connected to the coaxial microwave switch (Radiall R577 433002). The employed two-channel (channel A and channel B) switch is driven by a DC (V) bias. The factory defined RF continuity in the employed microwave switch for channel A is 5 → 8 and 7 → 6 (blue color in the switch cartoon), and for channel B is 5 → 6 and 7 → 8 (yellow color in the switch cartoon). It connects one of its input ports (output 4 in the figure) to the output read-out chain, and the same time it terminates the other input (output 3 in the figure) at 50 Ω impedance.

of generating small magnetic fields sufficient for qubit operations[46]. We have characterized the device using one tone and two tone-spectroscopy methods, briefly discussed later in this section, and in Supplementary Note 3.

## RF measurements

In this section, we report on one tone-spectroscopy measurements where we study the qubit-resonator interaction under small applied magnetic fields, and at various injected microwave powers. Transmission rectification manifests itself as the difference between the transmission coefficients $S_{31}$ and $S_{42}$. The transmission coefficient $S_{XY}$ is defined as the ratio of the signal amplitude coming out of the port X and of that going into the port Y. We tune the qubit to the degeneracy point, corresponding to half a flux quanta threading the qubit loop, $\Phi = 0.5\Phi_0$.

We sweep the probe signal frequency at different probing powers. To minimise possible errors caused by the attenuation in the input lines, we have applied an in-situ calibration method described in Section II under background calibration. In Fig. 4a, b we plot the transmission coefficient $|S_{31}|^2$ and $|S_{42}|^2$ obtained in this way.

We explore the range of frequencies around the frequency of the hybrid mode of the two resonators $f_h = 6.761$ GHz. This mode is formed because the resonators 1 and 2 are coupled not only to the qubit, but also to each other via the inductances $L_1, L_2$. At small microwave powers the frequency of this mode is shifted upwards dispersively induced by its coupling to the qubit, where $\chi = 22$ MHz, and the transmissions $S_{31}$ and $S_{42}$ exhibit resonances at the frequency $f_h + \chi = 6.784$ GHz. At high powers the hybrid mode decouples from the qubit and the resonance frequency moves from $f_h + \chi$ to $f_h$. At intermediate power levels the system shows strongly non-linear behavior

typical for the quantum Duffing oscillator[47–49]. The lowest power, required to drive the system in the strongly non-linear regime, is observed at resonance frequency $f_h + \chi$. We denote such power as $P_j^*$, where the index $j$ indicates the resonator through which the driving power is injected. In Fig. 4a we indicate the power $P_1^* = -112$ dBm by a solid black arrow, and in Fig. 4b we mark $P_2^* = -117$ dBm by a dotted black arrow. The observed power difference equals 5 dB, or 4.3 fW. It provides power scale of transmission rectification in our device.

We define the transmission rectification ratio ($R$)[50,51] as

$$R = \left| \frac{|S_{42}|^2 - |S_{31}|^2}{|S_{42}|^2 + |S_{31}|^2} \right| \qquad (1)$$

In Fig. 5a, d, g we plot this ratio as a function of frequency and magnetic flux for three different levels of the injected microwave power to show the increasing trend in transmission rectification ratio with the injected power.

We observe stronger transmission rectification close to the resonance frequencies corresponding to the hybrid modes of the resonators and the qubit. These modes are also revealed by the usual one tone-spectroscopy measurements, see Supplementary Fig. 2 in Supplementary Note 3. To illustrate the transmission rectification effect further, in Fig. 5b, c, e, f, h, i we plot the transmission coefficient $|S_{42}|^2$ and $|S_{31}|^2$ at the same three levels of the microwave power and at two selected values of the flux, $\Phi/\Phi_0 = 0.45$ and $\Phi/\Phi_0 = 0.5$. At the lowest power, −134 dBm, the transmission rectification happens only very close to the resonance frequencies, see Fig. 5a–c.

At low input power −134 dBm, in our device, the signal-to-noise ratio (SNR) is relatively poor. Therefore to avoid the rectification contributed by the background noise we retain only such data points

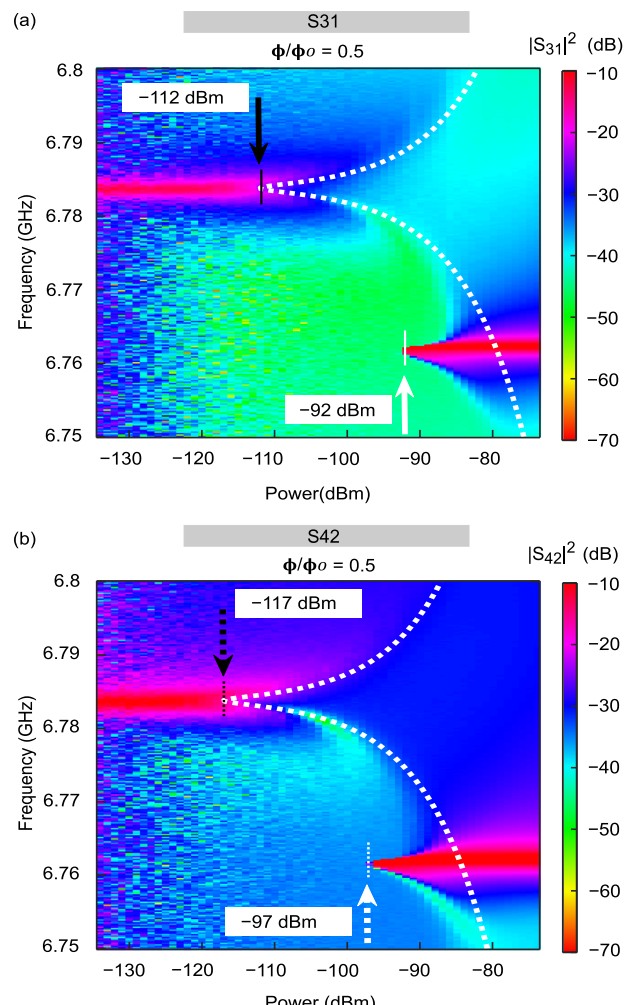

**Fig. 4 | Measured transmission coefficient of the device as a function of the injected microwave power and frequency. a** Transmission coefficient $|S_{31}|^2$. **b** Transmission coefficient $|S_{42}|^2$. The color bars represent the transmission amplitude in *dB*. The dotted lines are the theoretically expected positions of the peak maxima given by Eq. (2). The observed maximum difference between the background transmission (without background calibration) is less than 10%, and can be considered as an error bar. See Background calibration under Section II for error analysis.

where the sum $|S_{42}|^2 + |S_{31}|^2$ exceeds certain threshold value provided in the label above the graphs. In Fig. 5a, this value is $>9.5 \times 10^{-3}$, it allows us to compute the rectification precisely for the signal-of-interest avoiding any contributions from the background noise. A similar plot for −134 dBm with the background noise is reported in Supplementary Fig. 3, in Supplementary Note 4. In contrast, at relatively higher powers −114 dBm and −99 dBm (see Fig. 5d, g) the transmission rectification ratio exceeds 60% in the wide band of ≈250 MHz near the avoided crossing points, and for −99 dBm input power it exceeds 90% for a 50 MHz wide range between 6.81 GHz and 6.86 GHz. The insertion loss and isolation figures inferred from Fig. 5 are shown in Table 1. The maximum isolation achieved in the reported device at half flux is 19 dB for −114 dBm input power, though the insertion loss in the device is relatively poor and can be improved. At the intermediate power −114 dBm, we observe splitting of the single line in the transmission coefficient $S_{42}$ into two, whereas the transmission coefficient $S_{31}$ still maintains one line as it does at low microwave powers (see Fig. 5f).

The origin of the diode effect in our device is the non-linearity of the qubit. For this reason, single Lorentzian lines in the transmission coefficients $|S_{31}|^2$ and $|S_{42}|^2$ centered at frequency $f_h + \chi$ split into two

lines above the threshold values of the input powers $P_1^*$ and $P_2^*$. In our sample $P_1^* > P_2^*$ (see Fig. 4). The positions of the two peaks after the splitting, i.e. for $P > P_j^*$, are given by the expression

$$f_{\pm}(P_j) = f_h + \chi \pm \frac{\sqrt{2}}{3}\frac{\kappa_h}{2\pi}\sqrt{\frac{P_j^{in}}{P_j^*} - 1}. \qquad (2)$$

Here $\kappa_h/(2\pi) = 1.1$ MHz is the line-width of the peak at low power. The derivation of Eq. (2) is provided in Supplementary Note 2. In Fig. 4 we indicate the peak positions (Eq. (2)) by white dashed lines. They agree well with the experimentally observed ones. In the experiment we observe the ratio $P_1^*/P_2^* = 3.2$, which corresponds to a 5 dB difference in power on the log-scale. A strong transmission rectification effect with $R \sim 1$ occurs at powers $P \gtrsim P_2^*/3$, at which the peaks in $|S_{31}|^2$ and $|S_{42}|^2$ overlap weakly. Theory model reported in Supplementary Note 2 predicts that the ratio $P_1^*/P_2^*$ scales as $P_1^*/P_2^* = f_1^4 \kappa_{h2}/f_2^4 \kappa_{h1}$, where $\kappa_{h1}$ and $\kappa_{h2}$ are the partial contributions to the damping rate of the hybrid mode due to the leakage of the energy via the capacitors $C_{K1}$ and $C_{K2}$. Therefore, one can further enhance the diode effect in our system by making these capacitors unequal. In addition, improving the coupling asymmetry by exploiting the length dependent inductance of the coupling element could boost the device performance. In our device, the frequency bandwidth is limited by the tunability of the coupled qubit-resonator assembly due to the fixed bare resonator frequency. Replacing the fixed frequency resonator with a frequency tunable resonator by adding one or more Josephson junction based SQUID could facilitate larger and tunable rectifying band-with, making it more versatile from application point of view[52,53]. However, due to its compact size the device remains easily scalable towards multiple readout channels by integrating several resonator-qubit-resonator systems with different read-out frequencies.

In summary, we propose a flux tunable on-chip microwave diode architecture. It is based on a superconducting flux qubit inductively coupled to two superconducting resonators. Using one tone-spectroscopy, tuning the qubit to the degeneracy point by applying the half-flux quantum to it, and performing two separate measurements with microwave signals coming either through resonator 1 or resonator 2, we obtain 5 dB difference between the powers needed to drive the qubit to the strongly non-linear regime. Furthermore, near the qubit-resonator avoided crossing region we observe high transmission rectification ratio $R > 90\%$ for a narrow frequency bandwidth of 50 MHz, and $R > 60\%$ for a wider bandwidth of 250 MHz, at −114 dBm and −99 dBm input microwave power. The transmission rectification ratio is flux tunable and maximizes at −99 dBm input microwave power. Based on the reported resonator-qubit-resonator geometry, the future goal is to realize tunable photonic quantum heat valve[5,40] and quantum heat rectifier[37,39], in the field of circuit quantum electrodynamics. Furthermore, the reported strong non-reciprocal behaviour remains relevant for potential applications in quantum information[5,31,32], microwave read-out components[16,20,23] and optomechanics[54,55].

## Methods

### Fabrication

The device was fabricated on a highly resistive 675 $\mu$m thick Si wafer covered by 30 nm thick dielectric layer of $Al_2O_3$, which has been made using atomic layer deposition (ALD). In the next step, a 200 nm thick layer of superconducting niobium (Nb) has been sputtered using DC magnetron sputtering technique. This step has been followed by patterning of the feed lines, resonators and ground-planes on a 300 nm thick positive electron beam resist. Post baking is done at 150 °C for 5 min, followed by the reactive ion etching (RIE) using $CF_4 + O_2$ chemistry. Post baking step allows the resist to improve the adhesion with the substrate thus minimizing the chances of etching of unexposed Nb parts. The ALD deposited $Al_2O_3$ layer acts as an etch-stop-layer, preventing etching into Si. In the second lithography step, a bi-

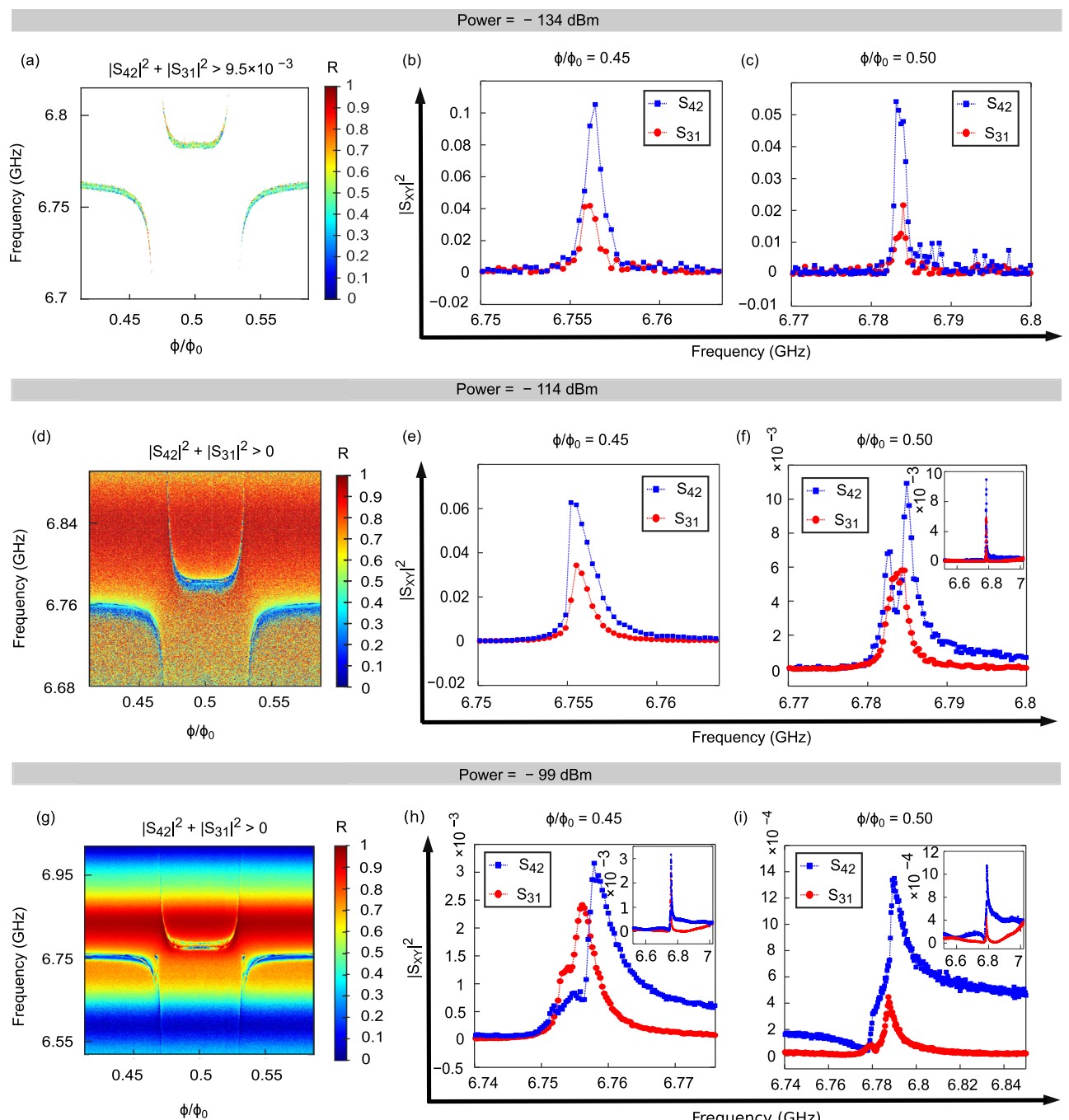

**Fig. 5 | Data analysis for estimation of the transmission rectification ratio $R$.**
**a**, **d**, **g** $R$ of Eq. (1) measured at three different levels of the injected microwave power ($P$), −134 dBm, −114 dBm and −99 dBm. The color bar in (**a**, **d**, **g**) is the rectification ratio $R$. Here, $\Phi$ is the external magnetic flux and $\Phi_0 (= h/2e)$ is the magnetic flux quantum. At low input power −134 dBm, see panel a, the SNR is relatively poor. Here we extract the data points where the sum $|S_{42}|^2 + |S_{31}|^2$ exceeds certain threshold value. This threshold value is provided in the label above the graphs. In this way we avoid errors emerging from the background noise that could

contribute to $R$ at low input power. (**b**, **e**, **h**) Transmission coefficient $|S_{31}|^2$ and $|S_{42}|^2$ for the same three levels of power and at the flux value $\Phi/\Phi_0 = 0.45$.
**c**, **f**, **i** Transmission coefficient $|S_{31}|^2$ and $|S_{42}|^2$ at the flux value $\Phi/\Phi_0 = 0.5$. The insets in (**f**, **h**, **i** are the plots with large x-axis from 6.52 to 7.02 GHz, of their main figures. The insertion loss and the isolation are reported in table I. Error bars for (**a**–**c**) are ± 35%, whereas for (**d**–**i**) the error bars are below ±10%. For more details on error estimation see error analysis under Section II.

layer PMMA/MMA resist has been used to write Josephson junctions using the standard Dolan bridge technique[56]. The exposed substrate has been developed in Methyl-Isobutyl-Ketone (MIBK): Isopropanol alcohol (IPA) developer solution, and Methyl-glycol-Methanol solution, respectively. Metal deposition has been performed using an e-beam evaporator. In-situ argon plasma milling has been performed on the sample surface to mill the native oxide before the evaporation.

It has been done to obtain a clean contact between Nb layers and Al layers. Afterwards, a 30 nm thick aluminium metal layer has been evaporated at +18° and it has been oxidized in-situ to create the tunnel barrier in the junctions. Subsequently, a second 30 nm thick aluminium layer is evaporated at −18°. To strip the deposited metal from the unexposed areas, samples are immersed in hot acetone for 40 min. The sample is visually examined under electron microscopy before

proceeding to dicing and measurements. The room temperature resistance of the 3 junction test SQUID fabricated on the same chip to mimic the real device has been measured to be $R \approx 2.6\,\mathrm{k\Omega}$.

## Error analysis and mitigation

**Background calibration.** In practice, the dominant errors could emerge due to the possible impedance mismatches at wire-bonds, PCB stage, microwave circuits, etc. and could lead to non-idealities which may have an effect on the symmetry and infinite waveguide assumption. Therefore to ensure the accuracy of the measurements, we perform an on-chip in-situ calibration as described below. We also note, that even without calibration the maximum observed difference between the background transmissions is less than 10%, while the observed difference in characteristic input powers $P_1^*$ and $P_2^*$ is over 50%.

Let us consider the measured transmission coefficient $S_{O_4I_1}$ from input 1 ($I_1$) to output 4 ($O_4$) (see Fig. 3b). It can can be expressed as

$$\tilde{S}_{O_4I_1} = \tilde{S}_{1I_1} + \tilde{S}_{41} + \tilde{S}_{O_44}. \tag{3}$$

Here $\tilde{S}_{ij} = 10\log_{10}(|S_{ij}|^2)$ is the transmission coefficient in decibel (dB), $\tilde{S}_{1I_1}$ is the attenuation factor from the input line 1 to the port 1 of the device, $\tilde{S}_{41}$ is the transmission coefficient between the ports 1 and 4, which we are looking for, and $\tilde{S}_{O_44}$ is the attenuation between the port 4 of the device and the input port of the microwave switch.

Similarly, we can write the transmission from the input 2 ($I_2$) to the output 3 ($O_3$) as

$$\tilde{S}_{O_3I_2} = \tilde{S}_{2I_2} + \tilde{S}_{32} + \tilde{S}_{O_33}, \tag{4}$$

**Table 1 | Device insertion loss and isolation estimated from Fig. 5 at 0.5 flux, for different input powers**

| S.No | Power (dBm) | Insertion Loss (dB) | | Isolation (dB) | |
|---|---|---|---|---|---|
| | | Resonance | Max. | Resonance | Max. |
| 1 | −134 | 9.6 | 9.6 | 4.9 | 4.9 |
| 2 | −114 | 16.6 | 29.4 | 5.3 | 19 |
| 3 | −99 | 25.7 | 30 | 3.1 | 15.7 |

Resonance corresponds to the values obtained at the resonance peak of the signal, and Max. refers to the maximum values of insertion loss and isolation achieved in the frequency range 6.5–7.02 GHz at $0.5\Phi_0$ flux. A 3 dB loss to the output signal is added to the reported insertion loss values. This loss is due to the equal splitting of the incoming photons from the resonators at the capacitive couplers to the feedline.

and the transmissions across the device as

$$\tilde{S}_{O_3I_1} = \tilde{S}_{1I_1} + \tilde{S}_{31} + \tilde{S}_{O_33}, \tag{5}$$

$$\tilde{S}_{O_4I_2} = \tilde{S}_{2I_2} + \tilde{S}_{42} + \tilde{S}_{O_44}. \tag{6}$$

We connect the ports 3 and 4 of the device to the input ports $O_3$ and $O_4$ of the microwave switch with the identical relatively short wires. The microwave switch specifications can be found in ref. 57. Therefore, we could assume that $\tilde{S}_{O_33} = \tilde{S}_{O_44}$.

Next, we perform spectroscopy at off-resonance frequencies at different flux points. Due to the strong qubit-resonator coupling the change in the dispersive shift under the applied magnetic flux is larger than the line-width of the resonance. Therefore, by sweeping the magnetic flux, we can tune the device to the off-resonance regime with $\tilde{S}_{41} = \tilde{S}_{32} = 0$ for each frequency and in this way can measure the background transmission coefficient $\tilde{S}_{O_4I_1\mathrm{bg}} = \tilde{S}_{1I_1} + \tilde{S}_{O_44}$ and $\tilde{S}_{O_3I_2\mathrm{bg}} = \tilde{S}_{2I_2} + \tilde{S}_{O_33}$ for the entire range of frequencies relevant for the experiment. They are plotted in Fig. 6a. Subtracting Eq. (3) from Eq. (5), Eq. (4) from Eq. (6), and recalling that $\tilde{S}_{O_33} = \tilde{S}_{O_44}$, we obtain the calibrated transmission coefficients of our system as

$$\tilde{S}_{31} = \tilde{S}_{O_3I_1} - \tilde{S}_{O_4I_1\mathrm{bg}}, \tag{7}$$

$$\tilde{S}_{42} = \tilde{S}_{O_4I_2} - \tilde{S}_{O_3I_2\mathrm{bg}}. \tag{8}$$

The same method has been used to calibrate the transmissions $S_{41}, S_{32}$.

The calibrated transmission coefficients obtained from Eqs. (7) and (8) at two different microwave powers are plotted in Fig. 6b, c. In Fig. 6c, corresponding to the high microwave power ($-74$ dBm) applied to the device at the magnetic flux $\Phi = 0.5\Phi_0$, we observe that $\tilde{S}_{31}$ and $\tilde{S}_{42}$ almost coincide in the frequency range from 4 GHz to 5.2 GHz, the difference between them does not exceed 1.5 dB. In contrast, in the range from 5.2 GHz to 7.8 GHz we observe significant difference, i.e. the diode effect, which reaches 15 dB near the frequency of the hybrid mode. Alternatively, using a two port device with a cryogenic circulator or a directional coupler at each input line one could also obtain the diode behaviour. Our choice of design allows a possibility to perform reflection measurements ($S_{41}, S_{32}$) and transmission measurements ($S_{31}, S_{42}$), providing a more comprehensive and complete knowledge of the reported system, useful for better understanding of the device and error mitigation.

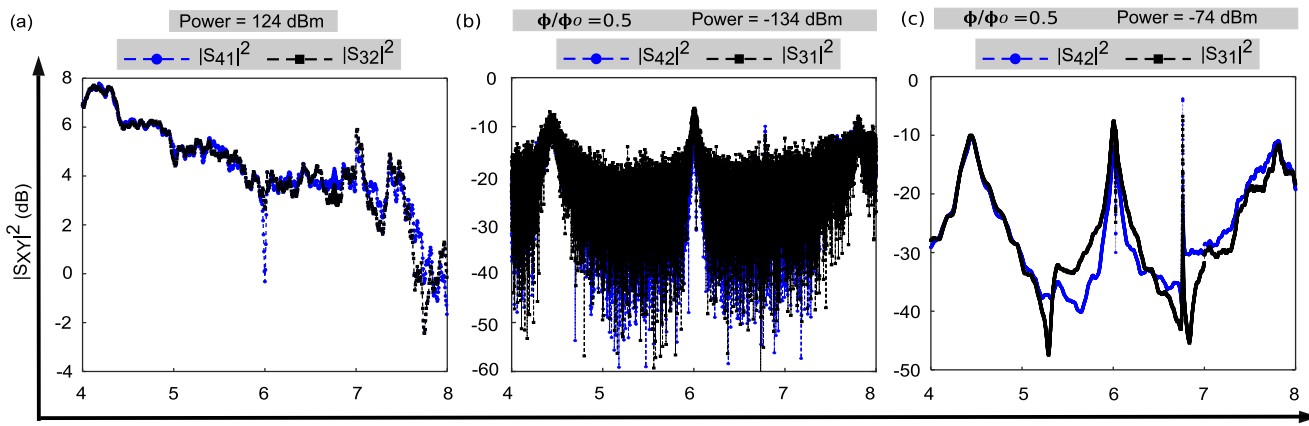

**Fig. 6 | Wide-band transmission coefficient graphs. a** Transmission coefficient plots for sides S41 and S32, reconstructed to obtain transmission baseline for background calibration. **b, c** Calibrated transmission coefficient plots obtained from Eqs. (7) and (8), measured across the device at −134 dBm and −74 dBm microwave power. The error bar is less than 10%, for more details see Background calibration under Section II.

**Error analysis at low input power, – 134 dBm.** For low input power, where the noise level is a sizable portion of the signal-of-interest, see Fig. 5b, c, we estimate the maximum error in $R$ due to the background noise in both directions. Our estimation reveals that the maximum error in $R$ due to this noise is $\approx 35\%$ for vanishing $R$. However, this contribution decreases at any non-zero value of $R$. For high powers, see Fig. 5e, f, h, i, the SNR is much higher as compared to that at the lowest input power. Therefore, the contribution in $R$ from the noise will be much less than 10%.

## Data availability
The data that support the plots within this article are available from the corresponding author upon request.

## Code availability
The codes that support the findings of this study are available from the corresponding author upon request.

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

## Acknowledgements
We acknowledge the financial support from Academy of Finland grants (grant number 297240, 312057 and 303677), and from the European Union's Horizon 2020 research and innovation programme under the European Research Council (ERC) programme (grant number 742559) and Marie Sklodowska-Curie actions (grant agreements 766025). We sincerely recognize the provision of facilities by Micronova Nanofabrication Centre, and OtaNano - Low Temperature Laboratory of Aalto University which is a part of European Microkelvin Platform EMP (grant number. 824109), to perform this research. We thank and acknowledge VTT Technical Research Center for provision of high quality sputtered Nb films.

## Author contributions
The conceptual idea was drafted by R.U. and G.T. The device design and fabrication was done by R.U. The measurements were carried by R.U. The data analysis is conducted by R.U., Y-C.C., D.S.G., and G.T. The theoretical model was conceived by D.S.G., in collaboration with G.T. The microwave simulations were carried by Y-C.C. and A.G. In this work D.S.G. and Y-C.C. contributed equally. The technical support was provided by J.T.P. The work was supported and supervised by J.P.P. The manuscript was written by R.U., with important contributions from all the authors.

## Competing interests
The authors declare no competing interests.
