## [Peer Review File · Nature Communications]

REVIEWER COMMENTS

Reviewer #1 (Remarks to the Author):

The authors report on a superconducting device consisting of two microwave resonators coupled through a flux qubit, an inductive path and a capacitive path (the latter is not explicitly mentioned in the manuscript). Both resonators are capacitively coupled, via nominally identically capacitors, to two feedlines, which are treated by the authors as infinite transmission lines.

Clear signs of nonreciprocity are reported in the manuscript. As motivated in the authors' introduction, quantum devices exhibiting nonreciprocal behaviour are attractive both from a practical and a fundamental point of view. The device studied by the authors relies on well established fabrication techniques and microwave technologies. This constitutes a considerable strength, since it could be potentially incorporated into quantum devices practically without any extra steps. Moreover, it does not require complicated time-domain protocols (i.e. it works on a continuous wave fashion), and does not seem to require careful and tedious tuning.

Below, I list some general comments and aspects concerning general aspects of the manuscript. In the next section, I list some more specific comments and questions pertaining to specific parts of the manuscript. Finally, I list some side comments and questions that came to mind while reading the manuscript.

General comments

- Non-reciprocity metrics: to quantify the degree of nonreciprocity, the authors use a figure of merit known as "rectification ratio" or "rectifying factor", which is commonplace in the quantum diode (theory) literature. More usual, however – and particularly in the context of microwave readout – are the figures of merit isolation and insertion loss. Rectification by itself is not a complete metric, since it fails to quantify insertion loss: for most applications it is not only critical to establish a one-directional photonic channel, but also to be able to transmit a significant fraction of the forward-field. Due to this, the figure of merit "diode efficiency" is also commonly quoted in the literature. Throughout the manuscript I found no mention to diode efficiency or insertion loss. These quantities can be inferred from the plots in Fig. 5, but I would consider useful to find them directly quoted. Although not optimal, they are relevant to the device and make it easy to the reader to understand the limitations and potential improvements on the device.

- Physical understanding of the device: in the place where I would expect a general intuition on how the device works (lines 67-70), I found a statement about qubit energy levels which was hard for me to relate to non-reciprocity on the device. Perhaps the authors can elaborate on this. Line 219 mentions the nonlinearity, which is, from a theory point of view (Lorenz reciprocity theorem), the only reason why the device can behave nonreciprocally. However, I think this does not help with the physical understanding of this particular device. From my point of view, the most interesting information is provided on Appendix B. From there, can I understand nonreciprocity of the device as a result of the nonlinear saturation of the flux qubit? Since the resonators are detuned from each other, the flux qubit sees different powers when probed from different ends, and thus has different degrees of saturation, which then reflect on transmission through the device (perfect reflection at ~ 0 photons, perfect transmission at and above saturation rate of the flux qubit, which would be proportional to its coupling rate to the respective resonator). To this, one would have to add that the resonators are also asymmetrically coupled to the flux qubit. Is this necessary to achieve nonreciprocity in the device?

- Device design and the assumptions made regarding transmission coefficients: The authors implemented a 4-port device, but the theory corresponds to a 2-port device. The differences are already clear in Fig 1a vs Fig 1c. I was initially confused by this and would have appreciated some explanation pertaining to the design choices. Furthermore, the authors assumed in the main text that $S_{12} = S_{42}$ without clarification or justification. In the supplement there are some mentions to the fact that the resonators should ideally couple symmetrically to both ports in the feedline, but these also remained unjustified. In practice, however, impedance mismatches at wirebonds, PCB, microwave connectors, etc. lead to non-idealities which could then have an effect on the symmetry/infinite waveguide assumption. I doubt these would lead to the observed non-reciprocal behaviour of the reported device, but I do find that this should be explicitly considered and not left to the reader to assume. Why a 2-port device, with semi-infinite transmission lines coupled to each port, was not considered? A circulator or a directional coupler at each input would have provided the option of obtaining all measured quantities with a potentially simpler device.

Questions/comments/doubts related to specific sections

- Introduction: I find the comparison between superconducting devices and other two-level systems biased. While superconducting qubits are ubiquitous in current research, many other types of TLS are considered for quantum computation. At the current stage it is hard to say which is “the” most promising candidate (cf. lines 25-26).

- Line 48: these devices have been implemented and demonstrated, not only proposed.

- Line 49: what do the authors mean here by noble materials?

- Literature review: seems biased towards diode/isolator superconducting devices and does not give a clear picture of what was achieved before in the field of on-chip non-reciprocity -- notable exceptions are the Hall-effect circulator implementations, e.g. Mahoney PRX 2017, electromechanical implementations e.g. Barzanjeh Nat Comms 2017, and the vast literature regarding superconducting on-chip circulators.

- Line 96: estimated from design or measurement?
- Line 112: estimates of g_i ?
- Fig. 2e: the non-uniform background is Al₂O₃? What justifies the choice of depositing the ALD layer? Does this improve the quantum devices compared to bare Si? Are there no concerns regarding the non-uniform layer? In line 276 it seems to be implied that EBL is performed for large features? Or is this e.g. a DUV optical lithography process with PMMA?
- Lines 150 and ff: I would like to reiterate that I found this rather confusing. As mentioned above, the quantities that one would expect here are S₂₁ vs S₁₂ (and a two-port device).
- Lines 181 and ff: $P_{\{j\}^{nl}} = P_{\{j\}^*}$? The authors use the latter terminology afterwards and in the App.
- Line 194: I understand that the data might look noisy, but I would still be interested in looking at the full unfiltered spectrum.
- Lines 257-258: Isn't power dependence rather a weakness than a strength, at least when considering readout applications?
- Line 307: If I understand correctly, this would not be possible with the cryo switch depicted in Fig. 3.
- Fig 6a: Power = -124 dBm (also note general lack of typographic rigour distinguishing hyphens from minus signs).

Appendix

- Theory: Why not a Lindbladian or input treatment? Ref for B2?
- Line 69: g_{12} arises from the capacitive coupling between resonators? Or also includes the inductive part? How does it relate to the lumped element inductances and capacitances?

Extra comments

- Tunability of device: although the flux qubit can be tuned, the resonators are fixed in frequency. How does this influence the tunability of the device?
- Role of dissipation/coherence in the device: no qubit losses considered. How do they influence the device? Would T₁ times somehow influence the low power behaviour of the device? Were the coherence times of the device quantified?
- What is the expected nonreciprocal time domain behaviour? In my understanding, saturation of the qubit would be necessary to reach the non-reciprocal regime. Therefore, at initial times the device would behave symmetrically.
- Double driving/noise driving from opposite input: in a more realistic situation (eg. in readout applications), the device is driven from both sides – a coherent tone in the forward direction, and

thermal noise in the opposite direction. How would the device behave under such conditions. See also dynamical nonreciprocity (Shi, Yu and Fan. Nat. Phot. 2015)

- Role of nonlinearity: would any two-level system suffice to show nonreciprocity? How would a (less anharmonic) transmon qubit behave?
- I would have been interested in finding a theoretical expression quantifying the degree of rectification/diode efficiency. A simple expression, if feasible, would be instructive to interpret the relevant parameters at play.

Andrés Rosario Hamann

Reviewer #2 (Remarks to the Author):

The miniaturization of non-reciprocal microwave devices on-chip is an essential theme for the scale-up and evolution of quantum circuits and quantum processors. In addition, as the authors point out, it may also contribute to expanding the frontiers of quantum thermodynamics.

The authors measured an interesting irreversible transmittance (quantum diode effect) in the microwave region in two transmission line resonator systems capacitively coupled to a flux qubit, and reported detailed analysis results. The frequency region that shows diode-like unidirectional transmittance is near the qubit-resonator avoided crossing region, which is about 50 to 250 MHz. The report is sound, very carefully tested and correctly analyzed. However, due to the many conditions imposed to achieve unidirectional transmittance (quantum diode effect), the actual use may be limited. Although it is possible to incorporate a quantum diode effect element into a quantum circuit chip, the operating temperature of the superconducting flux qubit (about 10 mK) is required, and the flux bias near the optimum operating point is also required. It would be necessary to apply a global magnetic field using coils to each of the integrated elements. There remain challenges that hinder scalability requiring magnets or magnetic fields.

The following points need to be properly addressed before making a decision.

[1] Similar work using flux qubits was reported by R. Navarathna et al. (Phys. Rev. Lett. 130, 037001 (2023).), and this paper should be appropriately cited.

Recently, a demonstration of the principle of a miniaturized on-chip microwave irreversible element that does not require a magnetic field in a wide band over the GHz frequency range by combining two SIS mixers has been reported (S. Masui et al., IEEE Microwave and Wireless Technology Letters, vol.33, no.7,

pp. 1051–1054, July 2023).

It is necessary to compare the merits and demerits of these methods at appropriate points in the paper, and to accurately inform the readers of the paper of the latest trends in this field.

[2] Equation (2) is important in the analysis of measurement results. Details of the derivation of this formula are given in Appendix B. Is the coupling g_{12} between the two resonators shown in H_{res} equation (B2) due to direct coupling? If the coupling is caused by a higher-order process due to the interaction between the flux qubit described in H_{int} equation (B4) and the two resonators, the suspicion of double counting arises. The authors should clearly indicate this point. I can't find a clear description even after reading the text and appendix.

We thank the reviewers for very useful and detailed comments on our work and the editor for favourable judgement. Below please find our responses to the reviewers comments and questions. The changes in the revised files and our response are marked with dark-blue color text. Thank you.

Reviewer #1 (Remarks to the Author):

The authors report on a superconducting device consisting of two microwave resonators coupled through a flux qubit, an inductive path and a capacitive path (the latter is not explicitly mentioned in the manuscript). Both resonators are capacitively coupled, via nominally identically capacitors, to two feedlines, which are treated by the authors as infinite transmission lines.

Clear signs of nonreciprocity are reported in the manuscript. As motivated in the authors' introduction, quantum devices exhibiting nonreciprocal behaviour are attractive both from a practical and a fundamental point of view. The device studied by the authors relies on well established fabrication techniques and microwave technologies. This constitutes a considerable strength, since it could be potentially incorporated into quantum devices practically without any extra steps. Moreover, it does not require complicated time-domain protocols (i.e. it works on a continuous wave fashion), and does not seem to require careful and tedious tuning.

Below, I list some general comments and aspects concerning general aspects of the manuscript. In the next section, I list some more specific comments and questions pertaining to specific parts of the manuscript. Finally, I list some side comments and questions that came to mind while reading the manuscript.

General comments

- Non-reciprocity metrics: to quantify the degree of nonreciprocity, the authors use a figure of merit known as “rectification ratio” or “rectifying factor”, which is commonplace in the quantum diode (theory) literature. More usual, however – and particularly in the context of microwave readout – are the figures of merit isolation and insertion loss. Rectification by itself is not a complete metric, since it fails to quantify insertion loss: for most applications it is not only critical to establish a one-directional photonic channel, but also to be able to transmit a significant fraction of the forward-field. Due to this, the figure of merit “diode efficiency” is also commonly quoted in the literature. Throughout the manuscript I found no mention to diode efficiency or insertion loss. These quantities can be inferred from the plots in Fig. 5, but I would consider useful to find them directly quoted. Although not optimal, they are relevant to the device and make it easy to the reader to understand the limitations and potential improvements on the device.

Our reply

Based on the referee recommendation, we have introduced the term ‘insertion loss and isolation ratio’ in the revised manuscript. This could be found in lines 45-49, 262-267, and briefly in the caption of Fig. 5. Also, we have summarized these in Table I in page 4 of the revised manuscript. We thank the referee for this recommendation. We have also added potential improvements on the device, mentioned in lines 293-308 in the revised manuscript.

- Physical understanding of the device: in the place where I would expect a general intuition on how the device works (lines 67-70), I found a statement about qubit energy levels which was hard for me to relate to non-reciprocity on the device. Perhaps the authors can elaborate on this. Line 219 mentions the nonlinearity, which is, from a theory point of view (Lorenz reciprocity theorem), the only reason why the device can behave nonreciprocally. However, I think this does not help with the physical understanding of this particular device. From my point of view, the most interesting information is provided on Appendix B. From there, can I understand nonreciprocity of the device as a result of the nonlinear saturation of the flux qubit? Since the resonators are detuned from each other, the flux qubit sees different powers when probed from different ends, and thus has different degrees of saturation, which then reflect on transmission through the device (perfect reflection at 0 photons, perfect transmission at and above saturation rate of the flux qubit, which would be proportional to its coupling rate to the respective resonator). To this, one would have to add that the resonators are also asymmetrically coupled to the flux qubit. Is this necessary to achieve nonreciprocity in the device?

Our reply

The referee is right, the non-reciprocal properties of the flux qubit arise from its non-linearity in combination with the asymmetric coupling to the two input ports. Because of this, different input powers should be applied via the two ports to excite the qubit to the same level. Following the recommendation of the referee we have added few sentences describing these properties in the revised version of the manuscript. We have placed these sentences in the Introduction, lines 87-96.

- Device design and the assumptions made regarding transmission coefficients: The authors implemented a 4-port device, but the theory corresponds to a 2-port device. The differences are already clear in Fig 1a vs Fig 1c. I was initially confused by this and would have appreciated some explanation pertaining to the design choices.

Furthermore, the authors assumed in the main text that $S_{12} = S_{42}$ without clarification or justification. In the supplement there are some mentions to the fact that the resonators should *ideally* couple symmetrically to both ports in the feedline, but these also remained unjustified. In practice, however, impedance mismatches at wirebonds, PCB, microwave connectors, etc. lead to non-idealities which could then have an effect on the symmetry/infinite waveguide assumption. I doubt these would lead to the observed non-reciprocal behaviour of the reported device, but I do find that this should be explicitly considered and not left to the reader to assume. Why a 2-port device, with semi-infinite transmission lines coupled to each port, was not considered? A circulator or a directional coupler at each input would have provided the option of obtaining all measured quantities with a potentially simpler device.

Our reply

The text in Fig 1 (a) has been modified for better understanding and now complies with Fig 1c, in both figures a 4 port device is reported. Our theory model presented in Appendix B has been developed for the 4-port system. In particular, Eqs. (B14) and (B18), which have used to analyze and fit the data, are valid for the 4-port device.

Regarding the assumption ($S_{12} = S_{42}$), we have updated the manuscript with the justification, please see lines 148-156. Thanks to the referee for bringing this to our attention.

As suggested by referee, in lines 371-374 (section IV), we have now explicitly mentioned the possibilities of impedance mismatch due to several experimental factors.

Referee is right, indeed a 2-port device could be an option too, and one could also obtain diode behaviour using a circulator or a directional coupler at each input line. Our choice of design gives us freedom to perform reflection measurements (i.e S_{41} and S_{32}), and transmission measurements (i.e S_{31} and S_{42}), hence providing a more complete information on the reported device. This information helped us to make more careful, thorough and accurate analysis reported in the manuscript, for example the background calibration method described in Section IV. We have addressed the possibility to use a two-port device and our justification for a 4 port device in lines 426-434, in the revised manuscript.

Questions/comments/doubts related to specific sections

- Introduction: I find the comparison between superconducting devices and other two-level systems biased. While superconducting qubits are ubiquitous in current research, many other types of TLS are considered for quantum computation. At the current stage it is hard to say which is “the” most promising candidate (cf. lines 25-26).

Our reply

We have reformulated the first paragraph of the Introduction according to the referee’s views. We now make clear that other types of two-level systems have some advantages and can be used in the future devices.

- Line 48: these devices have been implemented and demonstrated, not only proposed.

Our reply

We have corrected this in the revised version, please see the rephrased introduction.

- Line 49: what do the authors mean here by noble materials?

Our reply

We admit that the word selection ‘noble materials’ was not appropriate, and we have corrected this in the revised version.

- Literature review: seems biased towards diode/isolator superconducting devices and does not give a clear picture of what was achieved before in the field of on-chip non-reciprocity – notable exceptions are the Hall-effect circulator implementations, e.g. Mahoney PRX 2017, electromechanical implementations e.g. Barzanjeh Nat Comms 2017, and the vast literature regarding superconducting on-chip circulators.

Our reply

As suggested by the referee, in the revised version we have cited some more literature in the field of on-chip reciprocity. We kindly ask referee to read the introduction in the revised manuscript.

- Line 96: estimated from design or measurement?

Our reply

The value of α ($= 0.632$) is estimated by fitting the spectroscopy data based on the model described in the Appendix C and has a good agreement with the experimentally observed (from SEM image of the junctions) value of α which is 0.6. We have updated this information in the revised version for readers to better understand in lines 118-123.

- Line 112: estimates of g_i ?

Our reply

In the experiment we have measured the couplings between the qubit and the high-frequency and the low-frequency modes of the resonators, $g_h/(2\pi) = 175$ MHz and $g_l/(2\pi) \approx 1$ MHz (see Appendix C). According to our theory model presented in Appendix B, these couplings are related to g_1 and g_2 as

$$\begin{aligned} g_h &= -g_1 \sqrt{\frac{\omega_1}{\omega_h}} \sin \theta + g_2 \sqrt{\frac{\omega_2}{\omega_h}} \cos \theta, \\ g_l &= g_1 \sqrt{\frac{\omega_1}{\omega_l}} \cos \theta + g_2 \sqrt{\frac{\omega_2}{\omega_l}} \sin \theta, \end{aligned} \quad (1)$$

where the angle θ is defined in Eq. (B11). From these expressions and using the parameters given in the Appendix B we estimate $g_1/(2\pi) \approx -89$ MHz and $g_2/(2\pi) \approx 155$ MHz. We notice that the absolute value of g_2 is approximately 2 times bigger than $|g_1|$ because by design the coupling inductors depicted in Fig. 1(c) are related as $L_2 \approx 2L_1$. g_1 and g_2 have opposite signs because the current flowing through L_1 and L_2 is added to the current through the resonator 2 and subtracted from the current in the resonator 1. In the revised version of the paper we provide the estimates of g_1 and g_2 . We also add the above equations and the estimates for g_1, g_2 to the revised Appendix. We also provide the values for g_1 and g_2 in the main text, lines 139-142.

- Fig. 2e: the non-uniform background is Al2O3? What justifies the choice of depositing the ALD layer? Does this improve the quantum devices compared to bare Si? Are there no concerns regarding the non-uniform layer? In line 276 it seems to be implied that EBL is performed for large features? Or is this e.g. a DUV optical lithography process with PMMA?

Our reply

In Fig. 2 (c) (e), the non-uniform background around the aluminium structures are due to the resist residues after the lift-off process. In practise, these can be removed substantially using a dry milling process (oxygen plasma descum) that we have adopted very recently.

For etching of the patterned Nb ground layer we use reactive ion etching using SF6 + Ar chemistry, which has a lower selectivity for Nb/Si pair of materials. Therefore we have to use Al2O3 as an etch stop layer. ALD grown Al2O3 is a good candidate from the fabrication point-of-view, because it is nearly impossible to etch using SF6 + Ar chemistry, and its growth technique is rather simple and provides good uniformity (of thickness).

Using Al2O3 makes the fabrication process simpler and robust, but comes with a trade-off. Since Al2O3 is an amorphous material, therefore it induces two-level system (TLS) defects, limiting the coherence time of superconducting qubits. From our experience (and from the vast literature available), the internal quality factor of superconducting resonators without Al2O3 layer (or any insulating layer) is higher by an order of magnitude. To address this trade-off, one can also use a thin Al layer deposited right after the Nb layer without breaking the vacuum (in-situ deposition). After patterning and etching of the Nb ground planes, the Al layer is removed using wet etching process. We are optimizing the latter step for its future integration to our process flow.

Yes, in the reported work EBL is performed for large features too. Optical lithography or laser patterning can also be used for large features.

- Lines 150 and ff: I would like to reiterate that I found this rather confusing. As mentioned above, the quantities that one would expect here are S21 vs S12 (and a two-port device).

Our reply

We hope that our reply to the last comment in general section has made it clear why we employ a four-port device, and compare S31 vs S42 instead of a two port device (S21 vs S12). Also, we have addressed this in lines 426-434, in the revised manuscript.

- Lines 181 and ff: $P_j^{nl} = P_j^*$? The authors use the latter terminology afterwards and in the App.

Our reply

Yes, P_j^{nl} and P_j^* are the same. In the revised version we use the notation P_j^* , in the main text (see lines 221-226, 276-278, 285, 287, 289, 290, 415), and in the appendix (lines 68, 120, 124, 133, 134, 188, 191, 197-198, 204). We thank the referee for pointing out this unnecessary symbol.

- Line 194: I understand that the data might look noisy, but I would still be interested in looking at the full unfiltered spectrum.

Our reply

We have updated Fig. 5 (d) with full unfiltered spectrum for input power -114 dBm. Also in section IV of appendix, we have added the full spectrum in Fig. D.1, for input power -134 dBm.

- Lines 257-258: Isn't power dependence rather a weakness than a strength, at least when considering readout applications?

Our reply

Referee is right, it is more convenient to know how much power is transmitted through input to output, and how much power can be reduced from the reverse direction without complicated calibration when the properties are linear. However, the nonlinear properties here provide some benefit in more sophisticated way for qubit measurements. In general, for qubit measurements the transmitted powers on the transmission line is typically less than -100 dBm (low powers). The insertion loss at such power is low while passing through from the input. The high insertion loss at high power could avoid the amplification saturation and perhaps any damage to JPA (Josephson parametric amplifier). The high isolation of around 20 dB at working frequency range observed at relatively higher input powers is favourable to avoid noise from the reverse path.

- Line 307: If I understand correctly, this would not be possible with the cryo switch depicted in Fig. 3.

Our reply

We believe that there is some misunderstanding caused due to insufficient explanation provided in the manuscript. We have tried to explain it better in the modified Fig. 3 and its caption in the revised version. In interest to provide more details to the referee and readers, we have cited the microwave specification file in lines 393-394 and in Ref [60].

- Fig 6a: Power = -124 dBm (also note general lack of typographic rigour distinguishing hyphens from minus signs).

Our reply

We thank referee for pointing out this typo. We have corrected it in the revised version.

Appendix

- Theory: Why not a Lindbladian or input treatment? Ref for B2?

Our reply

Our model is equivalent to the Lindblad theory and/or input-output theory. Initially, it is slightly more general, because no rotating wave approximation has been made at the beginning. For Eq. (B2) we have added the following reference: M. J. Collett and C. W. Gardiner, Phys. Rev. A **30**, 1386 (1984).

- Line 69: g_{12} arises from the capacitive coupling between resonators? Or also includes the inductive part? How does it relate to the lumped element inductances and capacitances?

Our reply

In general, both inductive and capacitive coupling contribute to g_{12} . However, we believe that in our system the inductive coupling dominates. Although it is very difficult to derive an exact expression for g_{12} , we have managed to derive an approximate formula, which should be valid at sufficiently weak coupling,

$$g_{12} \approx \frac{2}{\pi} \frac{\omega_1 \omega_2 L'_1 L'_2}{Z_0} \left(C_q \omega_1 \omega_2 + \frac{1}{L_q} \right).$$

Here we use the same notations for the lumped elements as in Fig. 7(a). With the parameters given in the Appendix A and in the main text, $L'_1 = 0.1$ nH, $L'_2 = 0.25$ nH, $L_q = 0.5$ nH, $C_q \approx 100$ fF, $\omega_1/(2\pi) = 6.5$ GHz, $\omega_2/(2\pi) = 7.5$ GHz, and $Z_0 = 50$ Ω , this expression gives $g_{12}/(2\pi) \approx 200$ MHz, which is not far from the estimate made in a different way in the Appendix B, namely $g_{12}/(2\pi) \approx 313$ MHz. With these parameters, the inductive coupling term $\propto 1/L_q$ is 10 times bigger than the capacitive one ($\propto C_q \omega_1 \omega_2$). Namely, we find $g_{12}^{(L_q)} = 195$ MHz and $g_{12}^{(C_q)} = 18$ MHz. Physically, the inductance L_q includes the kinetic inductance of upper part of the SQUID loop, which contains the three junctions, and the kinetic inductances of the two junctions with bigger critical currents. In the revised version of the Supplement we provide the expression for g_{12} as Eq. (B4).

Extra comments

- Tunability of device: although the flux qubit can be tuned, the resonators are fixed in frequency. How does this influence the tunability of the device?

Our reply

The bare resonator frequencies are fixed in the reported work, nevertheless the coupled qubit-resonator system is tunable within a range of frequency with the applied magnetic flux as reported in the article. Certainly, the fixed bare resonator frequencies hinder the tunability of the reported device and affects its versatility from application point of view. Experimentally, the reported fixed frequency resonator could be replaced by a tunable resonator. A fixed frequency resonator can be engineered as a tunable resonator by adding one or more SQUIDs. The Josephson junctions in the SQUID act as tunable non/linear inductor. A vast amount of literature (experimental and theoretical) on tunable resonators is available, some of it is cited in the revised version. Alternatively, to achieve multiple rectifying bandwidth one could add more resonator-qubit-resonator assemblies designed at different read-out frequencies: thanks to the compact size of our device. Thanks to the referee for raising this point, we have added few sentences in the revised manuscript regarding this in line 300-308.

- Role of dissipation/coherence in the device: no qubit losses considered. How do they influence the device? Would T1 times somehow influence the low power behaviour of the device? Were the coherence times of the device quantified?

Our reply

Indeed, we do not explicitly discuss the losses in the qubit. In theory, the inverse time $1/T_1$ should be added to the full damping rate k_h of the high frequency hybrid mode. We have clarified this point in the revised version of the Appendix after Eq. (B10). Unfortunately, in our setup we could not measure the times T_1 and T_2 . However, the fitting values for the damping rates given after Eq. (B11) we can estimate the total internal damping rate as $\kappa_{hi} = \kappa_h - \kappa_{h1} - \kappa_{h2} = 1.24$ MHz. Therefore, we can find the low boundary for the relaxation time of the qubit $T_1 \geq \kappa_{hi}^{-1} = 0.8$ μ s. This estimate agrees well with what has been measured for other qubits fabricated in our lab with the same technology.

- What is the expected nonreciprocal time domain behaviour? In my understanding, saturation of the qubit would be necessary to reach the non-reciprocal regime. Therefore, at initial times the device would behave symmetrically.

Our reply

Yes, to achieve strongly non-reciprocal regime the qubit should be saturated if the power is applied from one side, and it should remain non-saturated if the power comes from the other side. It takes some time to achieve this limit after the power is switched on. This time should be of the order of κ_{c1}^{-1} or κ_{c2}^{-1} .

- Double driving/noise driving from opposite input: in a more realistic situation (eg. in readout applications), the device is driven from both sides – a coherent tone in the forward direction, and thermal noise in the opposite direction. How would the device behave under such conditions. See also dynamical non-reciprocity (Shi, Yu and Fan. Nat. Phot. 2015)

Our reply

According to Eq. (B15), the normalized amplitudes of the two signals coming from the opposite sides are added. If the two signals are not synchronized, one should add the two normalized powers. To get the transmission coefficient $|S_{31}|^2$ in this regime, in Eq. (B18) one should make the replacement

$$\frac{P_1}{P_1^*} \rightarrow \frac{P_1}{P_1^*} + \frac{P_2}{P_2^*}.$$

In practice it means that one should ensure that $P_1^* \gg P_2^*$. In this case, for example, the noise coming from the side 2 with $P_2 > P_2^*$ will not go through, while the signal with $P_1 < P_1^*$ will be transmitted. Concerning the model used in the paper Shi, Yu and Fan. Nat. Phot. 2015, we believe that is not directly applicable to our device. This model is essentially classical, while our device relies on the quantum effect — splitting of the transmission peak of a two-level system into two peaks at sufficiently strong power. This effect is related to Rabi splitting. Concerning the limits of rectification in our device, they are evident. Indeed, at low powers the rectification is absent. Similarly, at very high powers, where the non-linearity of Josephson junctions becomes unimportant, the device again becomes reciprocal. In spite of this, we believe that one can still use our diode in the regime $P_1^* \gg P_2^*$, as explained above.

- Role of nonlinearity: would any two-level system suffice to show nonreciprocity? How would a (less anharmonic) transmon qubit behave?

Our reply

The non-linearity is essential for the device operation. It is characterized by the parameter K in Eq. (B15). Strong rectification is achieved in the regime $K \gg \kappa_h$. If transmon qubit is sufficiently non-linear to satisfy this condition, it can also be used in such device. We note, however, that K differs from the anharmonicity of the bare qubit, it is the parameter characterizing the whole setup "qubit + two resonators". Therefore, K is smaller than the qubit charging energy.

- I would have been interested in finding a theoretical expression quantifying the degree of rectification/diode efficiency. A simple expression, if feasible, would be instructive to interpret the relevant parameters at play.

Our reply

The simple expression characterizing the diode efficiency is given by Eq. (B13) of the Appendix. It quantifies the asymmetry in the system and shows that the rectification disappears in fully symmetric case with $\kappa_{h1} = \kappa_{h2}$ and $\omega_1 = \omega_2$. We cite this expression in the new version of the paper in the paragraph after Eq. (2).

Summary of changes: Reviewer # 1

We have addressed each comment in detail in the above response to the reviewer 1. Below is the summary of changes made in the revised main text and appendix.

- As recommended by the referee, we have introduced TABLE I in the main text on page 4. The table includes the insertion loss and isolation values of the reported device.
- A general intuition on how the device works is added in the introduction section.
- Figure 1 (a) has been modified which complies with Fig. 1 (c). An explanation about the choice of a 4-port device is added to the main text.
- Introduction has been substantially modified with more literature based on the research in the field of on-chip reciprocity.
- Estimates of g_i are added and new equations for couplings based on the theory model is introduced in the revised Appendix. Values of g_1 and g_2 are provided in the revised main text.
- Following the referees interest to see the unfiltered spectrum of Fig. 5, we have updated Fig 5 (d) with the full unfiltered spectrum and the unfiltered spectrum for Fig 5 (a) is added in Fig D.1 of the appendix.
- Possible ways towards device performance improvement has been added to the revised manuscript.

Reviewer #2 (Remarks to the Author):

The miniaturization of non-reciprocal microwave devices on-chip is an essential theme for the scale-up and evolution of quantum circuits and quantum processors. In addition, as the authors point out, it may also contribute to expanding the frontiers of quantum thermodynamics. The authors measured an interesting irreversible transmittance (quantum diode effect) in the microwave region in two transmission line resonator systems capacitively coupled to a flux qubit, and reported detailed analysis results. The frequency region that shows diode-like unidirectional transmittance is near the qubit-resonator avoided crossing region, which is about 50 to 250 MHz. The report is sound, very carefully tested and correctly analyzed. However, due to the many conditions imposed to achieve unidirectional transmittance (quantum diode effect), the actual use may be limited. Although it is possible to incorporate a quantum diode effect element into a quantum circuit chip, the operating temperature of the superconducting flux qubit (about 10 mK) is required, and the flux bias near the optimum operating point is also required. It would be necessary to apply a global magnetic field using coils to each of the integrated elements. There remain challenges that hinder scalability requiring magnets or magnetic fields.

Our reply

Referee concerns are sound. The device we report is not a final product, rather it is a 'proof of principle'. There is room for improvement in the device performance. We have acknowledged this in the revised manuscript and proposed the possible route towards boosting the device performance, see lines 293-308 in the revised manuscript. Nevertheless,

the operation temperature of any Josephson junction based superconducting quantum bits is in few tens of milli-kelvin range, which is moreover a general requirement for any experiment on superconducting qubits.

The following points need to be properly addressed before making a decision.

[1] Similar work using flux qubits was reported by R. Navarathna et al. (Phys. Rev. Lett. 130, 037001 (2023).), and this paper should be appropriately cited. Recently, a demonstration of the principle of a miniaturized on-chip microwave irreversible element that does not require a magnetic field in a wide band over the GHz frequency range by combining two SIS mixers has been reported (S. Masui et al., IEEE Microwave and Wireless Technology Letters, vol.33, no.7, pp. 1051–1054, July 2023). It is necessary to compare the merits and demerits of these methods at appropriate points in the paper, and to accurately inform the readers of the paper of the latest trends in this field.

Our reply

The necessity of global magnetic field is insignificant in the context of our device, one could easily integrate a on-chip proximity based current bias line to generate the magnetic fields sufficient to operate the qubit without hindering the scalability due to large size of magnets. We have added the latter statement to the manuscript in line 182-184, with the mentioned article. We have cited the referee’s suggested papers and other related articles in the introduction section of the revised manuscript to inform the readers of the latest research in this field, please see lines 45-72 in the revised manuscript. We thank the referee for bringing our attention to these very interesting papers.

[2] Equation (2) is important in the analysis of measurement results. Details of the derivation of this formula are given in Appendix B. Is the coupling g_{12} between the two resonators shown in H_{res} equation (B2) due to direct coupling? If the coupling is caused by a higher-order process due to the interaction between the flux qubit described in H_{int} equation (B4) and the two resonators, the suspicion of double counting arises. The authors should clearly indicate this point. I can’t find a clear description even after reading the text and appendix.

Our reply

Yes, g_{12} in Eq. (B2) is the direct coupling between the two resonators. It is rather big and exceeds the couplings g_1 and g_2 . Namely, we estimate $g_{12}/(2\pi) = 313$ MHz, $g_1/(2\pi) = -89$ MHz and $g_2/(2\pi) = 155$ MHz. The details behind these estimates are explained in the revised version of the Supplement and in the reply to Reviewer 1, see above. In the revised version of the Supplement we provide the approximate expression for g_{12} as Eq. (B4). We also make clear in the text that g_{12} describes the direct coupling between the resonators, which is not sensitive to the state of the qubit.

Summary of changes: Reviewer # 2

All recommended changes proposed by the reviewer 2 are made and summarised below.

- Introduction has been majorly revised. The interesting papers mentioned by the reviewer are added to the introduction section informing readers of the recent trends in this field.
- Values of g_1 , g_2 and g_{12} are provided in the revised text. In the revised supplement we added an approximate expression for g_{12} as Eq. (B4).

REVIEWERS' COMMENTS

Reviewer #1 (Remarks to the Author):

I thank the authors for their careful and very detailed reply. I think all my comments have been properly addressed and would recommend publication of the manuscript in its present form.

A few last (minor) comments on their reply:

- line 71: charge noise is presented as a no-go. Charge-sensitivity tends to be problematic, but it can also be addressed and mitigated successfully, depending on the parameters of the device.

- ALD layer fab: Thanks for the detailed fab explanation, it is now clear to me that the Al₂O₃ ALD layer is there to stop the overetch into Si. However, this might still not be clear for all readers of the manuscript in its current form, since it is also common practice to overetch into Si (which is typically not an issue, unless the overetch is extreme and junctions cannot climb into the Nb base layer).

- Noise driving from input / dynamical nonreciprocity behaviour: I think the model in Eq. B15 might still not capture the effect of adding `_broadband_` thermal noise. But this was a side remark in any case, which I think becomes more relevant in a practical application of the device (e.g. for replacing circulators/isolators in the detection chain).

Andrés Rosario Hamann

Reviewer #2 (Remarks to the Author):

The authors responded appropriately to my comments and suggestions to improve the manuscript. Therefore, we recommend that the revised manuscript be published in Nature Communications, as it is recognized as a content that appropriately and accurately conveys cutting-edge research results to general readers.

We thank reviewers for evaluating our work and accepting it for publication in Nature Communications. We also deeply thank the editor for favourable judgment on the manuscript. Please find below our comments to the remaining remarks from reviewers in blue fonts.

REVIEWERS' COMMENTS

Reviewer #1 (Remarks to the Author):

- I thank the authors for their careful and very detailed reply. I think all my comments have been properly addressed and would recommend publication of the manuscript in its present form.

Our reply

We again thank the reviewer for reading our detailed response to the important and valuable comments. We believe that the reviewers comments and suggestions helped us to improve our manuscript. We are grateful for the reviewers decision to publish our work in Nature Communications.

A few last (minor) comments on their reply:

- line 71: charge noise is presented as a no-go. Charge-sensitivity tends to be problematic, but it can also be addressed and mitigated successfully, depending on the parameters of the device.

Our reply

Indeed, referee is right. Charge-sensitivity can be mitigated by carefully tuning the device parameters. We have mentioned this in the new version, please see lines 70-72.

- ALD layer fab: Thanks for the detailed fab explanation, it is now clear to me that the Al₂O₃ ALD layer is there to stop the overetch into Si. However, this might still not be clear for all readers of the manuscript in its current form, since it is also common practice to overetch into Si (which is typically not an issue, unless the overetch is extreme and junctions cannot climb into the Nb base layer).

Our reply

Thanks, in the updated version we have mentioned Al₂O₃ layer as etching stop layer for better understanding , see lines 350-352 under Methods section .

- Noise driving from input / dynamical nonreciprocity behaviour: I think the model in Eq. B15 might still not capture the effect of adding broadband thermal noise. But this was a side remark in any case, which I think becomes more relevant in a practical application of the device (e.g. for replacing circulators/isolators in the detection chain).

Our reply

We agree, the effect of strong broad-band noise may not be fully captured by Eq. (15). Indeed, this equation assumes that the driving signal is weak, and it does not consider the filtering effect of the resonators. However, we believe that Eq.(15) sufficiently well describes our experiment and it provides a simple physical picture of the rectification effect.

Reviewer #2 (Remarks to the Author):

- The authors responded appropriately to my comments and suggestions to improve the manuscript. Therefore, we recommend that the revised manuscript be published in Nature Communications, as it is recognized as a content that appropriately and accurately conveys cutting-edge research results to general readers.

Our reply

We thank the reviewer for the valuable suggestions that helped us to improve our manuscript. We thank the reviewer to recognize the value of our work and to recommend its publication in Nature Communications.